# Exploring parents' views of the use of narratives to promote childhood vaccination online

Eve Dubé[1,2]*, Marie-Eve Trottier[2], Dominique Gagnon[2], Julie A. Bettinger[3], Devon Greyson[3], Janice Graham[4], Noni E. MacDonald[4], Shannon E. MacDonald[5], Samantha B. Meyer[6], Holly O. Witteman[7], S. Michelle Driedger[8]

1 Department of Anthropology, Laval University, Quebec, Quebec, Canada, 2 Department of Biohazard, Quebec National Institute of Public Health, Quebec, Quebec, Canada, 3 Department of Pediatrics, Vaccine Evaluation Center, BC Children's Hospital Research Institute, University of British Columbia, Vancouver, British Columbia, Canada, 4 Department of Pediatrics, Dalhousie University, Halifax, Nova Scotia, Canada, 5 Faculty of Nursing, University, University of Alberta, Edmonton, Alberta, Canada, 6 School of Public Health Sciences, University of Waterloo, Waterloo, Ontario, Canada, 7 Department of Family Medicine, Laval University, Quebec, Quebec, Canada, 8 Department of Community Health Sciences, University of Manitoba, Winnipeg, Manitoba, Canada

* eve.dube@ant.ulaval.ca

**Data Availability Statement:** All relevant data are within the paper and its Supporting Information files.

## Abstract

### Background

Negative information about vaccines that spreads online may contribute to parents' vaccine hesitancy or refusal. Studies have shown that false claims about vaccines that use emotive personal narratives are more likely to be shared and engaged with on social media than factual evidence-based public health messages. The aim of this study was to explore parents' views regarding the use of positive narratives to promote childhood vaccination.

### Methods

We identified three ∼4-minute video narratives from social media that counter frequent parental concerns about childhood vaccination: parents and informed decision-making (online misinformation about vaccines); a paediatrician's clinical experience with vaccine-preventable diseases (prevention of still existing diseases); and a mother's experience with vaccine-preventable disease (risks of the disease). Focus group discussions were held with parents of children aged 0 to 5 years to assess their views on these three narratives and their general opinion on the use of narratives as a vaccine promotion intervention.

### Results

Four focus groups discussions were virtually held with 15 parents in December 2021. In general, parents trusted both health care provider's and parent's narratives, but participants identified more with stories having a parent as the main character. Both narratives featuring personal stories with vaccine-preventable diseases were preferred by parents, while the story about informed decision-making was perceived as less influential. Parents expressed

**Funding:** This research was funded by the Canadian Institutes of Health Research Catalyst Grant. ED received the grant. https://cihr-irsc.gc.ca/f/193.html The funders had no role in study design, data collection and analysis, decision to publish, or preparation of the manuscript.

**Competing interests:** The authors have declared that no competing interests exist.

the need for reliable and nuanced information about vaccines and diseases and felt that a short video format featuring a story was an efficient vaccine promotion intervention. However, many mentioned that they generally are not watching such videos while navigating the Web.

## Conclusion

While vaccine-critical stories are widely shared online, evidence on how best public health could counter these messages remains scarce. The use of narratives to promote vaccination was well-perceived by parents. Future studies are needed to assess reach and impact of such an intervention.

## Introduction

In Canada and elsewhere, an increasing number of parents are choosing to delay or refuse some or all recommended vaccines for their children, leading to declining community protection and subsequent outbreaks of vaccine-preventable diseases [1–8]. The ubiquity of anti-vaccine discourse on the Internet, particularly on social media sites, is considered by many experts to be a key driver of vaccine hesitancy [9].

The Internet and social media have provided multiple avenues for parents to "talk" about their children's health with other parents and medical experts, but also with pseudo-scientists and alternative care providers. Stories shared online are often powerful and can impact parents' decision-making [10, 11]. Evidence suggests vaccine-related stories often present "anti-vaccine" views in more compelling ways than public health "pro-vaccine" messaging. It is also increasingly recognized that parental vaccination decisions can be affected by a small number of vaccination stories or narratives [10, 12–14].

A narrative is defined as "a representation of connected events and characters that has an identifiable structure, is bounded in space and time, and contains implicit or explicit messages about the topic being addressed" [15 p. 222]. Recent meta-analyses have concluded that narratives are more effective than traditional educational or informational messages at increasing the intention to adopt specific health behaviours [16–18]. Narrative messages are more persuasive because people become absorbed in the story and can identify with characters, reducing resistance to persuasion [18, 19]. However, little is known about how to optimally use stories as a public health strategy to promote vaccines [12, 20]. Two interventions using narratives about HPV and COVID-19 vaccinations have been recently evaluated and positively influenced vaccine attitudes and vaccine uptake [21, 22].

The aim of this project was to explore parents' views on the use of narratives to promote childhood vaccination.

## Materials and methods

This article presents the findings of the first qualitative phase of a larger study aiming to develop and test narratives promoting childhood vaccination.

### Identification of narratives

First, we investigated different social media platforms (Facebook, Twitter, Instagram, TikTok, forums) and conducted Google searches to identify narratives about childhood vaccines

**Table 1. Recurrent themes in childhood vaccines narratives.**

| Position on vaccination | Themes |
|---|---|
| Negative narratives | • Vaccine safety concerns (e.g., vaccine can cause the disease or other diseases like epilepsy or sudden-infant death syndrome)<br>• False claims about the immune system (e.g., the immune system of the baby is too weak to take multiple vaccines)<br>• Vaccines perceived as unnecessary or not effective<br>• Conspiracy theories (e.g., big pharma) |
| Neutral narratives | • Vaccines perceived as useful for some people, but not everyone<br>• Balance of risks: vaccines have positive and negative effects |
| Positive narratives | • Positive experience with vaccination<br>• Vaccines perceived as useful<br>• Negative experience with vaccine-preventable diseases |

(videos and written posts) that Canadian parents could view online. We then conducted a content analysis among this corpus of online stories to identify patterns. Our analysis focused on creating a 'story map' of recurring topics, beliefs about vaccination (e.g., 'vaccines cause autism', 'vaccines are made for profit', 'vaccine-preventable diseases are mild', etc.), characters involved (e.g., parents, friends, partners, children, doctors, nurses, public health experts, pharmaceutical companies, etc.), events (e.g., medical consultation, illness episode, discussion with relatives, etc.) and type of message (e.g., warning, question, support, etc.). This exploratory phase allowed us to identify frequent parental concerns about childhood vaccination shared online (Table 1). Every narrative had a parent (mostly mothers) or an expert (e.g., health care provider, researcher) as the main character.

Subsequently, for expediency an online search was conducted to identify narratives (in video format) that counter the most recurrent negative themes that were identified, as described in Table 1. Furthermore, we searched for videos that were public and available on YouTube and Facebook in both French and English, the two official languages in Canada. For the paediatrician story video, we found a French video and an English video with a similar story. For the informed decision-making video featuring parents, the video was in English and we added French subtitles with the help of a bilingual PhD student and an undergraduate student. English sub titles were already available for the video of the mother's experience with vaccine preventable disease. Since the research team is bilingual, we did not need the help of a translator for data analysis or verbatim transcriptions. The story plot and main characters of these videos are described below (see S1 File for more information about the videos).

**Narrative #1: Parent's informed decision making [23].** A mother and a father shared that they did not get their first child vaccinated because others in their social network shared negative comments about vaccines, which made them hesitant. They later searched for more information about vaccines online and looked at both "anti-vaccination" websites and official sources (e.g., government, public health) and felt that official websites were more credible and trustworthy. The parents noted that, as they became better informed about childhood vaccinations, they made an informed decision and decided to have both their sons immunized as the best way to protect their health. The video ends with parents recommending looking for evidence-based information about vaccination and warning other parents about online misinformation.

**Narrative #2: A paediatrician's story [24, 25].** A paediatrician and infectious diseases specialist explain the importance of vaccines and their role in preventing many infectious diseases worldwide. He recounts difficult cases he had to handle with young children suffering

and eventually dying of vaccine-preventable diseases. The video ends with the paediatrician reminding of the importance of vaccinating children.

**Narrative #3: A mother's story [26].**   A mother is recounting the story of her son's illness and hospitalization due to a meningococcal disease (with a strain not covered by the available vaccines at that time). The video also features her child, now older, suffering from serious complications and permanently disabled. The mother shares the challenges of living with a child with severe disabilities. The video ends with a call to vaccinate by the mother, who wants to make sure that parents choose to vaccinate their children to avoid living the same difficult issues.

## Exploration of parents' views on the three narratives

Once narratives were selected, online focus groups discussions were held to assess parents' views on the three stories and to obtain general opinions regarding this type of vaccine promotion intervention (see S2 File for focus group guide).

**Recruitment.**   Participants were recruited through a sample of parents who participated in an online panel survey on COVID-19 pandemic and agreed to be recontacted for other studies in Québec, Canada [27]. We purposely excluded parents who expressed very positive attitudes about vaccines, as identified on an auto-evaluated 5-item vaccine hesitancy scale ("*Generally speaking, how hesitant about vaccines do you consider yourself to be*?"). Email invitations to participate in virtual focus groups discussion were sent to parents of at least one child under the age of 5, who considered themselves somewhat or very hesitant towards vaccination, who lived in Québec, Canada. *The repartition of English and French focus group discussion reflects Quebec's language demographic as the majority use French as their main language (86.3%). Although the majority of Quebecers speak French, we conducted focus group discussion with anglophones to explore potential cultural–linguistic differences.* To participate, parents needed to have access to a computer and a webcam and to be fluent in English or French.

**Data collection.**   A focus group guide was developed by the research team to assess parents' views about the three narratives (see Fig 1). During focus groups, each narrative was presented, followed by questions to assess parents' views on each video. The second part aimed to collect general comments on parents' perceptions of this type of interventions and suggestions about how to use such strategies as a public health intervention.

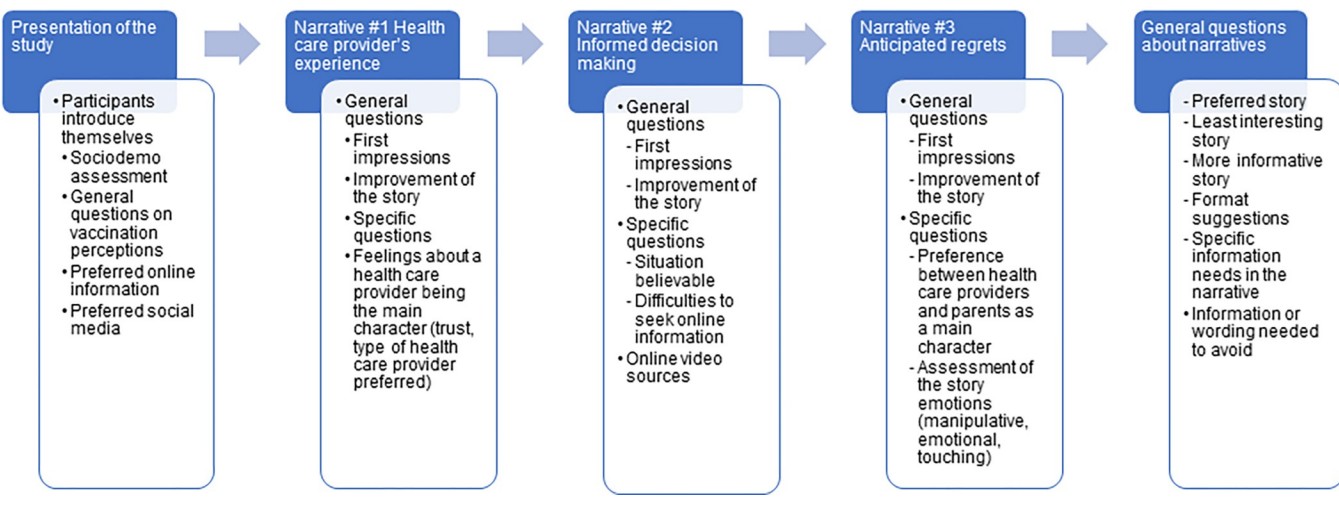

**Fig 1. Assessment of narratives during focus groups.**

All focus group discussions were recorded and transcribed verbatim. A deductive thematic analysis was conducted with NVivo12 using the main themes of the focus group guide as conceptual categories.

## Ethics

This study has been approved by the ethics committee of Centre Hospitalier Universitaire de Québec (2022–5680). Incentives of 50$ were given to each participant of the focus group for their time and involvement. An online informed consent form was signed before the focus groups.

## Results

In total, 101 invitations were sent to parents and 20 agreed to participate. Out of those, 5 dropped out after the reminder of the focus group because they were no longer available. Four focus groups with 3 to 5 participants each were conducted (Table 2). Level of hesitancy was determined according to self-evaluation using a 10-point scale. Three groups were conducted with French-speaking parents and one with English-speaking parents. Focus group length was between 45 and 90 minutes, depending on the number of people attending.

### Parents' opinions about childhood vaccination in general

Although we purposely excluded parents with very positive attitudes toward vaccines, the majority of participants had positive views about vaccination. Many participants were highly trusting of information about vaccines from healthcare providers and other medical sources (e.g. *"If the information comes from [name of a paediatric hospital], I would trust it completely"*). Most parents intended to or had already accepted routine vaccines for their child, and protection against infectious diseases was the most recurrent reason (e.g., *"When I think about vaccines, I think about protection and prevention"*). However, some parents did not feel that they need to vaccinate their children and expressed a preference for natural immunity (e.g. *"we have to let our kids develop their immune system by itself"*).

**Table 2. Focus group compositions.**

| Focus groups | Sex | Number of children under 5 years old | Level of hesitancy | Language |
|---|---|---|---|---|
| **#1** | Male | 1 | High | English |
| | Male | 2 | High | |
| | Male | 1 | High | |
| | Male | 1 | Low | |
| | Female | 1 | High | |
| **#2** | Male | 2 | Low | French |
| | Female | 2 | Low | |
| | Female | 1 | Low | |
| **#3** | Male | 3 | High | French |
| | Female | 2 | Low | |
| | Male | 1 | High | |
| | Male | 1 | Low | |
| **#4** | Female | 1 | Moderate | French |
| | Female | 1 | High | |
| | Male | 2 | Moderate | |

**Parents' views of the three narratives.** For the paediatrician's story, the majority of parents noted that when a doctor or a nurse talks about their experience, they usually trusted the information (e.g. *"I think the doctor is credible"*, *"no matter if it's a nurse or a doctor, I would listen to them because it's their speciality"*). However, this narrative was not well-perceived by the majority. This was mainly due to the context of the video (e.g. The paediatrician was outside of the clinic, in a park and provided information in a factual tone without much personal touch or emotions). Many participants noted that the health care provider wearing a "smock" or a "doctor's outfit" and including his professional affiliations (e.g., name of the association or the hospital) would have enhanced his credibility and increased trust.

Half of the parents liked the informed decision-making video featuring parents. These parents felt they could identify with other parents and shared similar concerns (e.g., *"I think the parents they do. . . provide a different perspective. They do show that's what they lived through, and they shared their own experiences. For sure, we can relate to them more as parents*). Others shared that they too have had difficulties to decide about vaccination for their children with all the contradictory information online. Parents who did not like the video were mainly thinking that having difficulties to find information online was not an important issue and noted that it's not that hard to find good information online (or to spot misinformation). Some parents also mentioned that a health care providers' call to vaccinate was likely to be more effective to move unsure parents toward vaccine acceptance (e.g., *"I think they also do need to introduce a doctor or a nurse in that video, maybe towards the end, in order to draw a conclusion or a shout out to action " get your kids vaccinated" or something, in the very end"*).

For the last video featuring a mother's experience with vaccine-preventable disease, the majority of participants identified with the mother and felt like the testimony was real, emotive and very powerful to move vaccine-hesitant parents (e.g., *"I think it is important to remind what vaccines are for because we don't see those diseases often"*,*"I think this video is sad, I think it shows the effect of the vaccines and I would vaccinate my child, yes"*). A few participants thought that it was too dramatic and played on guilt. These parents were worried about the potential for a backfire effect (e.g.," *I think there is a limit that should not be crossed (. . .) the only thing that is does is that it scares the population (. . .) if people against vaccination saw that, it would give them fuel to go more against it"*). In one of the focus groups, this situation created a debate. Some parents thought it was fair to show a video "playing on emotions" (i.e., guilt, fear) as this is how information is spread among "anti-vaccination" forums and websites whereas others considered that this was not the role of public health to do so. Table 3 summarizes the main themes that emerged from the focus group discussions for each narrative.

**Parents' opinions about use of narratives to promote vaccination.** In general, participants mentioned the need for more detailed and nuanced information to make a decision about their child's vaccination. Among information needs, parents mentioned information on brands and pharmaceutical information, vaccine preventable diseases and their consequences, nuanced information on the potential risks and benefits of vaccines. Many parents suggested adding information on sources of accurate information about vaccines (weblinks) at the end of the narratives.

All participants agreed that the narratives should be presented in a shorter video format (between ~20 seconds and one minute), but opinions about which social media platform to use were diverse (i.e., not one preferred one platform). In general, parents mainly mentioned using Facebook or YouTube to watch videos, but they mostly said that they did not watch those kinds of videos unless they are "forced" to, similar to a commercial on TV, radio or on the web before videos or between a music playlist (e.g., *"For me, it's sad to say, but I think the only way I would see that kind of video would be if it was a commercial that I have no choice to see. Because if it was on Facebook, I would not stop to look at it"*). Some parents suggested

**Table 3. Main themes for each narrative.**

| Narrative | Themes | Quotes |
|---|---|---|
| Parent's informed decision-making | • Accurate information<br>• Identified with parents' story<br>• Most nuanced and balanced video | *"Yeah, for sure, I can identify. With this vaccine for children that's coming on now, do I want to get my children vaccinated? Absolutely. But, right away? Maybe not. There are so many unknowns. "* |
| A paediatrician's story | Does not reflect reality of non-immunized children<br>• Do not feel concerned about the topic<br>• Need more information relating to risks and benefits about vaccination | *"During the video, we felt like the doctors was speaking more to other doctors. I think I am sharing his opinion and his perspectives, and how it impacted him. If you focus more on how it impacted him as a doctor, as a human being, as opposed to just. . . It felt really like directed to fellow doctors."* |
| A mother's experience with vaccine-preventable disease | • High credibility<br>• Acts on guilt<br>• Important information<br>• Identified with the mother | *"That's definitely the video that probably hits the most, closest to home, especially being a parent. I think that's a situation that nobody ever envisions being put in, as sad as it was to see the child in that state, I think that really could probably wake some people up in regards to the importance of all these types of childhood vaccines that are important to prevent the spread of illnesses."* |

presenting such narratives in waiting rooms of medical clinics to prompt further discussion about vaccination with healthcare providers. Others noted that these narratives could complement written information on governmental or hospital web pages. Parents felt that testimony from other parents were better because they could relate more to their experiences, but many felt that health care providers were more trustworthy. Finally, there were no strong opinions on the specific vaccine-preventable diseases that should be featured in the narratives, but some parents expressed that it would be more relevant, and of interest to them, if there was an epidemic or an outbreak of the specific disease at the moment they watched the video.

## Discussion

The use of emotive testimonies from parents of children that are believed to have been harmed by vaccination is a common tactic of vaccine-critical groups [28]. However, the use of similar approaches to promote vaccination is uncommon in health authorities' communication strategies due to concerns that this would be fearmongering and even unethical [29, 30]. Still, emerging evidence indicates that narratives are powerful communication tools that should be more widely used in public health communication interventions for behaviour change [18]. In this study, we explored perceptions of Canadian parents of young children about use of narratives to promote vaccination.

Findings of the focus groups' discussions indicate that parents are more likely to identify with other parents' stories. However, health care providers' advice and information about vaccines were perceived as more trustworthy. This is aligned with findings of other studies; the messenger is as important as the information that the narrative is trying to convey and trust in the main character(s) is needed to believe the story [31, 32]. During the discussion, parents also expressed the need for *balanced* or *nuanced* information (e.g., on the risks and benefits of vaccines, on vaccine-preventable diseases, on the role of the pharmaceutical industry, etc.), which is also a common request in other studies on vaccine acceptance [33, 34]. Although the search of nuanced information is frequently invoked by parents in order to make an informed decision about vaccines, it is still unclear what exactly type of information parents are

considering "nuanced", as often public health material is considered "too pro-vaccine" while vaccine critical websites are seen as "too anti-vaccine" [35].

While invoking negative emotions, guilt or fear in vaccine promotion interventions could create a backfire effect and be counter productive, most participants thought it was a good strategy to counterbalance vaccine critical stories that are shared online [36, 37]. In addition, parents all agreed that the video format on the web was the best way to reach them. The advantages of audio or video messages over print and written messages is well recognized in health communication literature [18, 31, 38]. However, most parents that participated in our study noted that they would be unlikely to watch such videos. Optimizing the benefits of using videos for vaccine promotion, it is of critical importance to ensure to have a captive audience when designing the intervention. It is still unclear in the literature if promoting vaccination through narratives has more impact when using negative (e.g., emphasis on the risk of vaccine-preventable diseases) or positive (e.g., emphasis on the safety, efficacy and usefulness of vaccines) framing. Some studies found that negative information is more convincing while others concluded that positive messages were more credible [36, 39, 40]. Findings of our qualitative study indicated that parents want to be informed about the benefits and risks about vaccines and found that messages focused only on benefits of vaccination were less credible and trustworthy [41, 42]. This indicates that narratives should not over emphasize the benefits of vaccines (e.g., no vaccine is 100% effective and 100% safe) and should include information about the potential risks of vaccines (e.g., mild, and common adverse events after immunization, rare risk of severe adverse events).

## Limitations

This qualitative study is exploratory, and findings may not be generalizable to other contexts. In addition, we were not able to achieve data saturation with regards to the preferred narrative. This may be because we were not able to interview all recruited participants, as some did not show up for the online focus group sessions, even if two email reminders were sent. As for all qualitative studies, desirability bias was also inevitable. Some participants that had a high level of vaccine hesitancy in the survey expressed more positive views about vaccines during the focus group discussion. The fact that most participants were positive about vaccination and that majority of their children were already immunized is a major limitation to understand views of hesitant parents about the videos. Two participants overtly expressed vaccine hesitancy during the focus group discussions and their opinions on the narratives were different. It is thus not possible to make comparisons with the views of other participants with more positive opinions about vaccination. In addition, we were not able to recruit many very hesitant parents. Perhaps these parents are less willing to participate in a study that aimed to develop positive messages about vaccines. Recruitment via social media using a question to filter non-hesitant parents has been shown to be an effective way of recruiting vaccine hesitant parents [43]. Finally, during the first two focus group discussions, the first narrative was the most disliked. To limit this bias, we rotated the order of presentation of the narratives for the last two focus group discussions.

## Conclusion

Childhood vaccination is a thorny issue that poses "wicked" risk communication problems for public health authorities. Wicked problems are by their very nature persistent and hard-to-resolve because they do not lend themselves to a scientific consensus about the best means for resolving the problem [44]. While scientific consensus on the public health benefits of vaccination are unequivocal, there is no such agreement on how best to use communication to

respond and guide efforts to address vaccine hesitancy. With the public's growing use of social media to inform health decisions, it is vital for public health experts to understand these platforms and how they could be mobilized for vaccine promotion interventions [45]. Although recent studies have shown positive impact of debunking myths on vaccination acceptance, these approaches can backfire for the highly hesitant [46, 47]. By promoting messages on the importance of vaccination to prevent disease instead of focusing on specific myths, the use of narratives is a promising approach. In this qualitative study, the use of narratives to promote vaccination was well-perceived by parents. As some participants suggested, it could be beneficial for online narratives in a video format to feature on governmental pages about vaccine information to make a decision. This could facilitate dissemination of narratives-based vaccine promotion strategies. The feasibility of this avenue should be explored as past analysis have highlighted that official websites are generally not using such approaches [42]. Larger quantitative studies are needed to assess the reach and impact of such an intervention.

## Supporting information

**S1 File.**
(DOCX)

**S2 File.**
(DOCX)

**S3 File.**
(DOCX)

## Acknowledgments

We would like to thank Mélissa Picard-Filiatrault biomedical undergraduate and Laurie-Ann Carrier, anthropology student, who helped us with the exploratory phase and during focus groups.

## Author Contributions

**Conceptualization:** Julie A. Bettinger, Devon Greyson, Janice Graham, Noni E. MacDonald, Shannon E. MacDonald, Samantha B. Meyer, Holly O. Witteman, S. Michelle Driedger.

**Formal analysis:** Marie-Eve Trottier.

**Funding acquisition:** Eve Dubé.

**Investigation:** Marie-Eve Trottier, Dominique Gagnon.

**Methodology:** Eve Dubé.

**Writing – original draft:** Marie-Eve Trottier.

**Writing – review & editing:** Eve Dubé, Dominique Gagnon, Julie A. Bettinger, Devon Greyson, Janice Graham, Noni E. MacDonald, Shannon E. MacDonald, Samantha B. Meyer, Holly O. Witteman, S. Michelle Driedger.

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
