## [Decision Letter · Decision Letter 0]

28 Sep 2022

PONE-D-22-21909Exploring parents’ views of the use of narratives to promote childhood vaccination onlinePLOS ONE

Dear author(s). 

Thank you for submitting your manuscript to PLOS ONE. After careful consideration, we feel that it has merit but does not fully meet PLOS ONE’s publication criteria as it currently stands. Therefore, we invite you to submit a revised version of the manuscript that addresses the points raised during the review process.

ACADEMIC EDITOR:  

The research participants and/or authors who were responsible for content /video creation should be specified. Authors should also indicate if translations were made by professional translators. Please address the questions, comments and concerns from each reviewer. 

We look forward to receiving your revised manuscript.

Kind regards,

Asrat Genet Amnie, MD, EdD, MPH, MBA

Academic Editor

PLOS ONE

Journal Requirements:

"We would like to thank Mélissa Picard-Filiatrault biomedical undergraduate and Laurie-Ann Carrier, anthropology student, who helped us with the exploratory phase and during focus groups and the Canadian Institutes of Health Research for funding the study."

"This research was funded by the Canadian Institutes of Health Research Catalyst Grant. ED received the grant. https://cihr-irsc.gc.ca/f/193.html The funders had no role in study design, data collection and analysis, decision to publish, or preparation of the manuscript."

4. PLOS requires an ORCID iD for the corresponding author in Editorial Manager on papers submitted after December 6th, 2016. Please ensure that you have an ORCID iD and that it is validated in Editorial Manager. To do this, go to ‘Update my Information’ (in the upper left-hand corner of the main menu), and click on the Fetch/Validate link next to the ORCID field. This will take you to the ORCID site and allow you to create a new iD or authenticate a pre-existing iD in Editorial Manager. Please see the following video for instructions on linking an ORCID iD to your Editorial Manager account: https://www.youtube.com/watch?v=_xcclfuvtxQ.

Additional Editor Comments:

The research participants and/or authors who were responsible for content /video creation should be specified. Authors should also indicate if translations were made by professional translators. Please address all the questions, comments and concerns from each reviewer.

Reviewers' comments:

Reviewer's Responses to Questions

**Comments to the Author**

1. Is the manuscript technically sound, and do the data support the conclusions?

Reviewer #1: Yes

Reviewer #2: Yes

Reviewer #3: Yes

2. Has the statistical analysis been performed appropriately and rigorously? 

Reviewer #1: N/A

Reviewer #2: N/A

Reviewer #3: N/A

3. Have the authors made all data underlying the findings in their manuscript fully available?

Reviewer #1: No

Reviewer #2: No

Reviewer #3: Yes

4. Is the manuscript presented in an intelligible fashion and written in standard English?

Reviewer #1: Yes

Reviewer #2: Yes

Reviewer #3: Yes

5. Review Comments to the Author

Reviewer #1: This article reports the findings of focus groups to determine what parents think and feel about particular kinds of pro-immunisation narratives. It is well written, reports sound research, and reflects appropriate analysis. These suggestions are intended to improve the manuscript. The discussion of COVID 19 vaccines is a bit of a red herring, especially as most studies discussed are not in the context of childhood vaccination. Suggest focusing mainly on childhood vaccines as it's important to do this work regardless of COVID! The authors situate this study within a broader one. As such, I think they could emphasise that they chose existing narratives for expediency. The fact that they analysed these narratives to draw out commonalities and main themes before choosing exemplars is good. However, I think they also needed to consider where these videos were stored and saved and who made them. Might this matter to the viewers? Was this discussed in the FGs? If not, this is a limitation of the study. The other issue was about language. It wasn't evident that all videos were in both languages, even though the authors said they were. Please be clear and transparent about this. It seems that for videos to exist in both languages they'd need to be Canadian but this also wasn't self evident. Also, with some data collected in English and some in French, when was it translated, how, and by whom? At what point in the analysis? In the results, the authors could lean more into whether the particularly Vax hesitant people said different things to the other participants. Also the use of 'blouse' is weird - blouses are specifically a type of women's shirt. Do you mean shirt? When people, at line 223, think "this is not an important issue", what do you mean by "this"?

I think a really interesting finding from this study is that government probably needs to pay for the team's eventual narratives to be disseminated. People won't go looking for them. Suggest leaning into this more.

Reviewer #2: This is a well-written study about an important topic. I enjoyed reading the manuscript and learning about the authors’ findings. I have some minor suggestions to help improve readability and to clarify a few points in the paper.

Methods:

Table 1: Can the authors please clarify what the ‘n’s represent in this table? For example, for ‘negative narratives (n=26)’, does this indicate that 26 different negative stories were identified even if some of these stories appeared in different forms across different platforms (meaning maybe >26 unique videos, posts, articles were found with some duplicate content), OR does this indicate that 26 unique online content pieces were identified and a Facebook post about a specific myth would be counted separately from a Tik Tok video with a creator repeating that same myth. I recognize this is a nuanced question. I think understanding whether the study team viewed a total of 37 pieces of online content vs >37 pieces of content that all referenced back to about 37 narratives would help the reader contextualize the extent of the search used to identify narratives.

Data collection sub-section: I think the last sentence of this paragraph should refer to the focus group guide (not interview grid) being available in appendix 1. The interview grid does appear as Figure 1, just as the authors have stated in the first sentence of this paragraph. Suggest rewording, or if I have misinterpreted then I’d suggest the authors provide more clear definition of what is meant by ‘interview grid’ vs ‘focus group guide’

Minor comment: In the references, the dates for publication are in French whereas the rest of the text is in English. Suggest changing reference language format for consistency.

Results: Can the authors provide any context about having 3 focus groups with French-speaking parents and 1 with English-speaking parents? Is this generally representative of the population in Quebec? Or is it aligned with know demographics associated with vaccine hesitancy in Quebec?

Table 2: Please provide a legend or footnote indicating the meaning of the asterisk included after the heading ‘Level of hesitancy’.

‘Parents’ opinions about childhood vaccination in general’ subsection: This paragraph notes that participants had very positive attitudes toward vaccines, but Table 2 notes six parents at having ‘high’ vaccine hesitancy and others with moderate hesitancy. Can authors please provide some comment to reconcile these two findings? Is information in table derived from survey scores and information in the text based on qualitative responses?

Discussion

Third paragraph: The sentences that begin ‘This finding highlight that…’ and ‘Finally, the literature is still unclear in the literature’ are both difficult to read due to problems with subject/verb agreement or possibly typographic errors. I’d ask the authors or one of the editors to review these two sentences closely to revise and improve clarity/readability.

Reviewer #3: Qualitative study from Quebec that aims to address online vaccine misinformation employing three different short (approx. 4 min) video narratives on vaccine decision making by parents. The authors conducted online focus group discussions and found that parents appreciated the narratives. However, parents pointed out they would rarely watch such online videos during their vaccine decision making process. The authors correctly conclude that the impact of such video narratives remains unclear and similarly, the best approach to countering online vaccine misinformation remains uncertain. Despite the negative results, the study is well designed, addresses an important topic, the data is presented in a clear fashion and the manuscript is well written.

Specific comments

1) As stated by the authors (line 197), most participants held positive views of vaccines, and (line 201) intended to or already had vaccinated their children. This is a limitation that should be mentioned more clearly in the discussion and/or limitations section.

2) In addition to mentioning the inability to recruit vaccine-hesitant parents to the study, authors should (in the discussion) provide a few sentences about how previous researchers went about recruiting vaccine-hesitant parents ( = the target group of the proposed online narratives), and what approach the authors suggest in future studies to reach/recruit vaccine-hesitant parents.

3) Debunking false vaccine information by health care providers and public health authorities has repeatedly been shown to be ineffective in changing vaccine-hesitant parents minds. In light of this, authors should cite these negative studies and discuss how such online narratives might be adapted to better reach vaccine-hesitant parents.

6. PLOS authors have the option to publish the peer review history of their article (what does this mean?). If published, this will include your full peer review and any attached files.

Reviewer #1: No

Reviewer #2: No

Reviewer #3: No

---

## [Author Response · Author response to Decision Letter 0]

9 Nov 2022

Dear editor, 

We thank the reviewer for their comments and feedback. Please find below our responses and the corresponding changes. All changes in the manuscript were done using the track change mode and a clean version is also submitted. We have made some changes in the manuscript according to your comments and the revisors comments. We also made minor clarifications in the manuscript to ensure readers comprehension. See below our answers to the comments. 

ACADEMIC EDITOR: 

1. The research participants and/or authors who were responsible for content /video creation should be specified.

Answer: Videos used for this project were not created for our specific project but were identified through the first phase of our project. The videos were all publicly available on YouTube and Facebook. For the paediatrician story: Arnaud Gagneur from Sherbrooke University for the French version ( https://www.youtube.com/watch?v=B_kpGPZeShI) and The Children’s Hospital of Philadelphia for the English version (https://www.youtube.com/watch?v=xNbVjCdvrdA ). For the informed decision-making videos featuring parents: Kids Boost Immunity (https://www.facebook.com/iboostimmunity/posts/1432957773404880/). For the mother’s experience with vaccine preventable disease: Immunize Canada (https://www.youtube.com/watch?v=mWPZGYAdcnc ). 

We added information in the methods about the videos (The links to the videos used during focus group discussion are available in supporting information 1)

2. Authors should also indicate if translations were made by professional translators. Please address the questions, comments and concerns from each reviewer. 

Answer: We added this information on line 139: For the paediatrician story video, we found a French video and an English video with a similar story. For the informed decision-making video featuring parents, the original video was in English and we added French subtitles with the help of a bilingual trainee. English subtitles were already available for the video of the mother’s experience with vaccine preventable disease. For data collection, focus groups were conducted in both French and English and transcribed in original language. The analysis was conducted with verbatim in their original language. All quotes in French were translated by the authors for the manuscript.

3. We note that you have provided funding information that is not currently declared in your Funding Statement. However, funding information should not appear in the Acknowledgments section or other areas of your manuscript. We will only publish funding information present in the Funding Statement section of the online submission form. Please remove any funding-related text from the manuscript and let us know how you would like to update your Funding Statement. Currently, your Funding Statement reads as follows: 

"This research was funded by the Canadian Institutes of Health Research Catalyst Grant. ED received the grant. https://cihr-irsc.gc.ca/f/193.html The funders had no role in study design, data collection and analysis, decision to publish, or preparation of the manuscript." Please include your amended statements within your cover letter; we will change the online submission form on your behalf.

Answer: We removed the funding statement in the acknowledgments section and the funding section. This should go in the funding section: “This research was funded by the Canadian Institutes of Health Research Catalyst Grant. ED received the grant. https://cihr-irsc.gc.ca/f/193.html The funders had no role in study design, data collection and analysis, decision to publish, or preparation of the manuscript."

Answer: After consultation of our institution research ethics board, we submitted a minimal anonymized data set as a Supporting information file 3.

4. PLOS requires an ORCID iD for the corresponding author in Editorial Manager on papers submitted after December 6th, 2016. Please ensure that you have an ORCID iD and that it is validated in Editorial Manager. To do this, go to ‘Update my Information’ (in the upper left-hand corner of the main menu), and click on the Fetch/Validate link next to the ORCID field. This will take you to the ORCID site and allow you to create a new iD or authenticate a pre-existing iD in Editorial Manager. Please see the following video for instructions on linking an ORCID iD to your Editorial Manager account: https://www.youtube.com/watch?v=_xcclfuvtxQ.

Answer: we added the ORCID ID.

Your ethics statement should only appear in the Methods section of your manuscript. If your ethics statement is written in any section besides the Methods, please move it to the Methods section and delete it from any other section. Please ensure that your ethics statement is included in your manuscript, as the ethics statement entered into the online submission form will not be published alongside your manuscript. 

Answer: We have put the ethics section in the methods. 

Answer: We added the information on line 173:” (see supporting information 2 for interview guide).” 

Answer: We reviewed the reference list. 

7. Reviewer#1 The discussion of COVID 19 vaccines is a bit of a red herring, especially as most studies discussed are not in the context of childhood vaccination. Suggest focusing mainly on childhood vaccines as it's important to do this work regardless of COVID! The authors situate this study within a broader one. 

Answer: We agree with the reviewer’s comment and our project started prior to the pandemic. Sentences related to COVID-19 pandemic and vaccines were removed to avoid confusion.

8. As such, I think they could emphasise that they chose existing narratives for expediency. The fact that they analysed these narratives to draw out commonalities and main themes before choosing exemplars is good. However, I think they also needed to consider where these videos were stored and saved and who made them. Might this matter to the viewers? Was this discussed in the FGs? If not, this is a limitation of the study. 

Answer: As noted in response to the Editor’s comment Videos used for this project were not created for our specific project but were identified through the first phase of our project. The videos were all publicly available on YouTube and Facebook. For the paediatrician story: Arnaud Gagneur from Sherbrooke University for the French version (https://www.youtube.com/watch?v=B_kpGPZeShI ) and The Children’s Hospital of Philadelphia for the English version (https://www.youtube.com/watch?v=xNbVjCdvrdA ). For the informed decision-making videos featuring parents: Kids Boost Immunity (https://www.facebook.com/iboostimmunity/posts/1432957773404880/ ). For the mother’s experience with vaccine preventable disease: Immunize Canada (https://www.youtube.com/watch?v=mWPZGYAdcnc ). 

We added information in the methods about the videos (The links to the videos used during focus group discussion are available in supporting information 1)

We also added on line 136: “Subsequently, for expediency an online search was conducted to identify narratives (in video format) that counter the most recurrent negative themes that were identified, as described in Table 1.”

9. The other issue was about language. It wasn't evident that all videos were in both languages, even though the authors said they were. Please be clear and transparent about this. It seems that for videos to exist in both languages they'd need to be Canadian but this also wasn't self evident. Also, with some data collected in English and some in French, when was it translated, how, and by whom? At what point in the analysis?

Answer: As noted in response to the Editor’s comment: We added this information on line 138: For the paediatrician story video, we found a French video and an English video with a similar story. For the informed decision-making video featuring parents, the video was in English and we added French subtitles with the help of a bilingual undergraduate student English subtitles were already available for the video of the mother’s experience with vaccine preventable disease. For data collection, focus groups were conducted in both French and English and transcribed in original language. The analysis was conducted with verbatim in their original language. All quotes in French were translated by the authors for the manuscript.

10. In the results, the authors could lean more into whether the particularly Vax hesitant people said different things to the other participants. 

Answer: Only two participants overtly expressed their hesitation toward vaccination during the focus groups and their opinions on the videos were different (i.e., one participant really appreciated the mother’s video while the other disliked it). We added a sentence in the limitations.

11. Also the use of 'blouse' is weird - blouses are specifically a type of women's shirt. Do you mean shirt? 

Answer: We changed the word “blouse” for “smock” as we were referring to doctors and nurses usual wearings.

12. When people, at line 223, think "this is not an important issue", what do you mean by "this"?

Answer: We changed “this” for “having difficulties to find information online”

13. I think a really interesting finding from this study is that government probably needs to pay for the team's eventual narratives to be disseminated. People won't go looking for them. Suggest leaning into this more.

Answer: Line 357 we added: “As some participants suggested, it could be beneficial for online narratives in a video format to feature on governmental pages about vaccine information to make a decision. This could facilitate dissemination of narratives-based vaccine promotion strategies. The feasibility of this avenue should be explored as past analysis have highlighted that official websites are generally not using such approaches.”

Reviewer #2:

14. Table 1: Can the authors please clarify what the ‘n’s represent in this table? For example, for ‘negative narratives (n=26)’, does this indicate that 26 different negative stories were identified even if some of these stories appeared in different forms across different platforms (meaning maybe >26 unique videos, posts, articles were found with some duplicate content), OR does this indicate that 26 unique online content pieces were identified and a Facebook post about a specific myth would be counted separately from a Tik Tok video with a creator repeating that same myth. I recognize this is a nuanced question. I think understanding whether the study team viewed a total of 37 pieces of online content vs >37 pieces of content that all referenced back to about 37 narratives would help the reader contextualize the extent of the search used to identify narratives.

Answer: The n referred to the number of videos that were analyzed for each category (negative, neutral, positive). We agree with reviewer’s comment that this information does not add much to the manuscript and have removed the n of the table. 

Data collection sub-section: I think the last sentence of this paragraph should refer to the focus group guide (not interview grid) being available in appendix 1. The interview grid does appear as Figure 1, just as the authors have stated in the first sentence of this paragraph. Suggest rewording, or if I have misinterpreted then I’d suggest the authors provide more clear definition of what is meant by ‘interview grid’ vs ‘focus group guide’

Answer: We changed interview grid for focus group guide everywhere in the manuscript. We added on line 173: (Focus group guide is available in supporting information 2)

15. Minor comment: In the references, the dates for publication are in French whereas the rest of the text is in English. Suggest changing reference language format for consistency.

Answer: We have reviewed all the references to make sure it is up to date and in English. 

16. Results: Can the authors provide any context about having 3 focus groups with French-speaking parents and 1 with English-speaking parents? Is this generally representative of the population in Quebec? Or is it aligned with know demographics associated with vaccine hesitancy in Quebec?

Answer: The majority of Quebecers are having French as their first language and 13.7% English. We added a sentence in the Methods line 182: “The repartition of English and French focus group discussion reflects Quebec’s language demographic as the majority use French as their main language (86.3%). Although the majority of Quebecers speak French, we conducted focus group discussion with anglophones to explore potential cultural – linguistic differences.”

17. Table 2: Please provide a legend or footnote indicating the meaning of the asterisk included after the heading ‘Level of hesitancy’.

Answer: We removed the asterisk and clarified the hesitancy scale in the main text line 209: “Level of hesitancy was determined according to self-evaluation using a 10-point scale. “

18. ‘Parents’ opinions about childhood vaccination in general’ subsection: This paragraph notes that participants had very positive attitudes toward vaccines, but Table 2 notes six parents at having ‘high’ vaccine hesitancy and others with moderate hesitancy. Can authors please provide some comment to reconcile these two findings? Is information in table derived from survey scores and information in the text based on qualitative responses?

Answer: Line 218 we changed “all” for “many”.

19. Discussion

Third paragraph: The sentences that begin ‘This finding highlight that…’ and ‘Finally, the literature is still unclear in the literature’ are both difficult to read due to problems with subject/verb agreement or possibly typographic errors. I’d ask the authors or one of the editors to review these two sentences closely to revise and improve clarity/readability.

Answer: Line 314: We changed “this finding highlight that” for “Optimizing the benefits of using videos for vaccine promotion, it is of critical importance to ensure to have a captive audience when designing the intervention.” Line 303: we removed: “the literature is still unclear” to avoid repetitions.

Reviewer #3:

20. As stated by the authors (line 197), most participants held positive views of vaccines, and (line 201) intended to or already had vaccinated their children. This is a limitation that should be mentioned more clearly in the discussion and/or limitations section.

Answer: We added on line 338: “The fact that most participants were positive about vaccination and that majority of their children were already immunized is a major limitation to understand views of hesitant parents about the videos.”

21. In addition to mentioning the inability to recruit vaccine-hesitant parents to the study, authors should (in the discussion) provide a few sentences about how previous researchers went about recruiting vaccine-hesitant parents ( = the target group of the proposed online narratives), and what approach the authors suggest in future studies to reach/recruit vaccine-hesitant parents.

Answer: We added this sentence in the limitations, line 342: Recruitment via social medias using a question to filter non-hesitant parents has been shown to be an effective way of recruiting vaccine hesitant parents(43). 

22. Debunking false vaccine information by health care providers and public health authorities has repeatedly been shown to be ineffective in changing vaccine-hesitant parents minds. In light of this, authors should cite these negative studies and discuss how such online narratives might be adapted to better reach vaccine-hesitant parents.

Answer: We added precisions, line 354: Although recent studies have shown positive impact of debunking myths on vaccination acceptance, these approaches can backfire for the highly hesitant (43,44). By promoting messages on the importance of vaccination to prevent disease instead of focusing on specific myths, the use of narratives is a promising approach.

---

## [Decision Letter · Decision Letter 1]

23 Mar 2023

Exploring parents’ views of the use of narratives to promote childhood vaccination online

PONE-D-22-21909R1

Dear Author (s):

We’re pleased to inform you that your manuscript has been judged scientifically suitable for publication and will be formally accepted for publication once it meets all outstanding technical requirements.

Kind regards,

Asrat Genet Amnie, MD, EdD, MPH, MBA

Academic Editor

PLOS ONE

Additional Editor Comments (optional):

Please provide point-by-point response to the reviewers' comments and and make sure to incorporate them  in the final version of your manuscript.

Reviewers' comments:

Reviewer's Responses to Questions

**Comments to the Author**

1. If the authors have adequately addressed your comments raised in a previous round of review and you feel that this manuscript is now acceptable for publication, you may indicate that here to bypass the “Comments to the Author” section, enter your conflict of interest statement in the “Confidential to Editor” section, and submit your "Accept" recommendation.

Reviewer #4: (No Response)

Reviewer #5: (No Response)

2. Is the manuscript technically sound, and do the data support the conclusions?

Reviewer #4: Partly

Reviewer #5: Yes

3. Has the statistical analysis been performed appropriately and rigorously? 

Reviewer #4: N/A

Reviewer #5: N/A

4. Have the authors made all data underlying the findings in their manuscript fully available?

Reviewer #4: Yes

Reviewer #5: Yes

5. Is the manuscript presented in an intelligible fashion and written in standard English?

Reviewer #4: Yes

Reviewer #5: Yes

6. Review Comments to the Author

Reviewer #4: Many thanks for submitting an interesting exploratory qualitative study, which I have peer reviewed with interest. Overall the paper is written well and presents some interesting findings on the acceptability of online video narratives for informing parents about childhood vaccinations. I believe addressing the following queries would help improve the paper prior to publication:

1. Throughout the paper you talk about ‘narratives’, which is a little ambiguous. I wonder whether it would be worth rephrasing this to ‘online video narratives’, as this is a clearer description of the social media content that the authors reviewed with participating parents.

2. Lines 11-12: The authors have provided a very succinct aim for the study, which leaves the reader a little unclear about what elements of parents’ views were to be explored. I would suggest expanding the aim to provided a bit more information about what the authors planned to explore e.g., beliefs, perceptions, anticipated risks, motivators, barrier, previous social media experience, etc.

3. Line 119: Did you include YouTube in your social media search for ‘online video narratives’ and if not why was this platform excluded? Since YouTube is arguably the largest online video sharing platform, I would suggest that exclusion of this data source during the narrative identification exercise is a significant limitation that needs to be addressed in the Discussion.

4. Line 134: The table number has been dropped here and needs to be readded to clarify the authors are discussing Table 1.

5. Methods – general comment: I would suggest that the authors review the COREQ guidelines, which provides a checklist for presenting qualitative research from qualitative interviews or focus groups. For example, I would be keen to see more detailed information about the methods used, for example - research team and their reflexivity, relationship with participants, theoretical framework, participant sampling (see below), and data collection (see below).

6. Methods - Recruitment: The authors mention that focus group participants were recruited after completing a COVID-19 survey. It would be useful to know more about this recruitment process and any potential biases that may have informed the participants’ underlying perceptions or beliefs. What was the survey about? When was the survey conducted? How were parents recruited (e.g. inclusion and exclusion criteria)? Overall, I’m interested in hearing how the recruitment process may have influenced your participants’ opinions. I’m also keen to know how your focus group participants relate / represent members of the general public, so any background info would helpful.

7. Line 176 - Figure 1: The quality of this figure is a low poor and a higher resolution version would be required for the final publication.

8. Data collection: As outlined in the COREQ guidelines, I would like to know a bit more detail about how the data was collected. In particular, did the authors use a semi-structured focus group approach and were prompts used? Was video or audio recording used during the focus groups? Were transcripts returned to participants for comments?

9. Results – line 189: The authors state that 101 invitations were sent to parents, was this to 101 separate parents, or does this mean that multiple invites were sent to fewer parents? Also, do you know anything about the characteristics of those parents that declined to participate?

10. Table 2 – Male parent had a professional domain as ‘print’, can you clarify what this means for the reader, as this is a little vague.

11. Results – general comment: It might be useful to describe which focus group (#1, 2, 3 or 4) each of the quotes comes from or better yet, would it be possible to attribute each of the quotes to an individual participant via a unique ID number? At the moment it is difficult to tell if the quotes are all coming all from the same focus group, or indeed from the same person.

12. Line 226 – there is an extra space after ‘towards’.

13. Line 227 – there is an extra space before ‘get’

14. Line 235 – there is an extra space before ‘I think’

15. Table 3: I’m not convinced that the quotes provided in the final column relate to the ‘General opinions’ outlined in column 2. I would suggest included more appropriate quotes or reconsidering the design of this table

16. Table 3 – A paediatrics’ story: The general opinion column fails to mention ‘trust’, yet this is discussed in the main text – does this need to be added? Also, as outlined above, the quote seems a little our of context considering the general opinions listed in column 2.

17. Line 245: The authors describe that participants mentioned the need for more ‘nuanced information’, indeed this is also discussed in detail during the discussion section. However, the authors fail to present any empirical evidence to corroborate this claim – is there a quote(s) that demonstrate this wish?

18. Discussion section – line 278: The authors state that advice from HCPs was perceived as more trustworthy, but this is not included in the ‘general opinions’ in Table 3.

19. Line 288-290: Sorry, I’m a little unclear about what the authors are stating here, could you reword please?

20. Line 296: ‘…a captive audience’ – can you clarify what you mean?

21. Lines 296-7: The word ‘literature’ is duplicated.

22. Line 314: The authors need to add the word vaccine i.e., ‘very vaccine hesitant’.

23. Lines 317-8: I would suggest including information about rotating the videos in the methods section.

24. Conclusion – Line 320 & 321: The word ‘wicked’ is somewhat lost in translation to an international audience, as it can have multiple meanings (some good and some bad). For clarity, I would suggest amending to something less open to interpretation.

Good luck with the amendments and your future work in this area.

Reviewer #5: This manuscript presents the results from a focus group study where parents who reported some level of vaccine hesitancy were shown several pro-vaccine videos and asked to share their opinions and perspectives about the efficacy of the videos. I appreciate that the authors are adding detailed qualitative data to an area of research where this is lacking. And I can see that the authors have been responsive to a prior round of revisions. I have just a few minor suggestions that I feel would strengthen the manuscript:

- I appreciate the level of detail about the videos and the translation work that was added in this revision. I do feel it's important for the authors to discuss in the limitations section the fact that two videos featured subtitles for translation whereas the third video was actually two distinct videos - one in English and one in French - that covered the same topics but may have differed in substantive ways. Further, did the authors note any differences in focus group reactions to the English video vs the French video?

- I see that the authors responded to prior feedback about whether the videos were created for this study or not. They write that they used existing videos "for expediency," but I feel the authors could really discuss this as a strength of the manuscript. Using existing videos rather than videos created for research purposes helps lend credibility and external validity to the findings.

- Given the finding that many participants said they would not be likely to watch these videos if they encountered them in their social media feeds, and indeed said they would be more likely to watch them if they were "forced to" such as in a TV or radio advertisement, I was surprised to not see the authors focus on this as a conclusion of the study. To me the data suggest that these types of videos might indeed be more effectively delivered as TV or radio advertisements rather than as social media narratives. This is a noteworthy finding!

- Finally, while the manuscript is written in a clear and engaging way, there are several typographical and grammatical errors that should be fixed. For example, the authors refer to "social medias" several times, but media is already plural, and thus the correct term is "social media". There are several other instances of incorrect pluralization and subject-verb agreement throughout the manuscript. They do not interfere with readability, but given that this journal does not provide copy editing services, these errors should be addressed by the authors.

7. PLOS authors have the option to publish the peer review history of their article (what does this mean?). If published, this will include your full peer review and any attached files.

Reviewer #4: No

Reviewer #5: **Yes: **Fashina Aladé

---

## [Editor Report · Acceptance letter]

31 Mar 2023

PONE-D-22-21909R1 

Exploring parents’ views of the use of narratives to promote childhood vaccination online 

Dear Dr. Dubé:

I'm pleased to inform you that your manuscript has been deemed suitable for publication in PLOS ONE. Congratulations! Your manuscript is now with our production department. 

Kind regards, 

on behalf of

Dr. Asrat Genet Amnie 

Academic Editor

PLOS ONE